# Antagonism between PTP1B and PTK Mediates Adults’ Insulin-Like Signaling Regulation of Egg Diapause in the Migratory Locust

**DOI:** 10.3390/insects12030253

**Published:** 2021-03-17

**Authors:** Shuang Li, Jie Wang, Dongnan Cui, Kun Hao, Jun Chen, Xiongbing Tu, Zehua Zhang

**Affiliations:** 1State Key Laboratory for Biology of Plant Diseases and Insect Pests, Institute of Plant Protection, Chinese Academy of Agricultural Sciences, Beijing 100193, China; sclishuang61@163.com (S.L.); wangjie5403@126.com (J.W.); Cuidongnan88@163.com (D.C.); haokun8611@foxmail.com (K.H.); cjhp2014@163.com (J.C.); 2School of Resources and Environmental Sciences, Henan Institute of Science and Technology, Xinxiang 453003, China

**Keywords:** insulin-like signaling, PI3K, Akt, embryo diapause, insect, diapause related protein

## Abstract

**Simple Summary:**

It was reported that insulin-like and fork head transcription factor (FOXO) are involved in the regulation of diapause in insects. However, the upstream modulators of the insulin-like signaling pathway (ISP) involved in diapause regulation are still unknown. We used RNAi and an inhibitor to treat PTK and PTP1B in adult tissues and injected Prx V protein or RNAi Prx V under both short and long photoperiod conditions to identify both proteins and broader cellular metabolism influences on diapause regulation. We found that under short photoperiod conditions PTP1B in female adults induces egg diapause, whereas PTK in female adults inhibits egg diapause. Intriguingly, we also found that the antioxidant enzyme Prx V is a negative regulator of NADPH oxidizing reaction, and apparently decreases reactive oxygen species (ROS) production and NADPH-OX activity. Thus, these results indicate that PTP1B, PTK and Prx V are upstream modulators that regulate diapause in eggs via the insulin signaling pathway. Furthermore, these findings have revealed a possible bridge connecting diapause hormone signaling to the insulin-like signaling pathway.

**Abstract:**

Diapause is a physiological development arrest state that helps insects to adapt to seasonality and overcome adverse environmental conditions. Numerous reports have indicated that insulinlike and fork head transcription factor (FOXO) are involved in the regulation of diapause in insects. However, the upstream modulators of the insulin-like signaling pathway (ISP) involved in diapause regulation are still unknown. Here, we used RNAi and an inhibitor to treat PTK and PTP1B in adult tissues and injected Prx V or RNAi Prx V under both short and long photoperiod conditions and monitored effects on the expression of ISP genes, the phosphorylation levels for IR and IRS, the activity of NADPH oxidase, the accumulation of reactive oxygen species (ROS) and energy metabolism, seeking to identify both proteins and broader cellular metabolism influences on diapause regulation. We found that under short photoperiod conditions PTP1B in female adults induces egg diapause, whereas PTK in female adults inhibits egg diapause. Intriguingly, we also found that the antioxidant enzyme Prx V is a negative regulator of NADPH oxidizing reaction and apparently decreases ROS production and NADPH-OX activity. In contrast, all the eggs laid by adults that were treated with a series of knockdown or purified-protein injection experiments or inhibitor studies and that were reared under long photoperiod conditions hatched successfully. Thus, our results suggest a mechanism wherein diapause-related proteins (PTP1B, PTK, and Prx V) of female adults are the upstream modulators that regulate offspring eggs’ diapause process through the insulin-like signaling pathway under short photoperiod conditions.

## 1. Introduction

Diapause is a seasonal adaptation to the environment, widely existing in invertebrate taxa so that allows insects at different stages of development to respond to periodic environment changes. For example, *Bombyx mori* can enter diapause in the embryonic stage [1], *Diatraea grandiosella* can enter diapause in the larval stage [2], *Sarcophaga crassipalpis* can enter diapause in the pupal stage [3], and *Drosophila montana* exhibits adult reproductive diapause [4]. Regardless of which stage it is initiated in, the characteristics of diapause are similar: suppressed metabolic activity and arrested development [5].

It has been reported that diapause hormone (DH) [6,7], juvenile hormone (JH), and molting hormone (MH) [8], as well as prothoracic-tropic hormone (PTTH) [9] (pp. 353–412), are involved in the regulation of insect diapause. Hormone regulation is understood as the main tactic that insects use to induce diapause, often during the time before they encounter the harsh environmental conditions of winter. More recently, experiments with adult diapause in *Drosophila melanogaster* [10,11] and the mosquito *Culex pipiens* [12,13], as well as with pupal diapause in the flesh fly *Sarcophaga crassipalpis* [14], have provided strong evidence that insulin-like signaling is also an important component of the regulatory pathway controlling diapause. Studies of insulin-like signaling and of the fork head transcription factor (FOXO), a downstream molecule in the insulin-like signaling pathway, have led to insights about how insects regulate diapause during overwintering [15,16]. Subsequently, insulin-like signaling and FOXO have been confirmed as regulators of diapause in mosquito [17] and in fruit flies [18]. Thus, there is now a consensus that a shutdown of insulin-like signaling prompts the activation of the downstream gene FOXO, thereby leading to diapause phenotypes [17,18].

Furthering our understanding of the upstream regulatory factors that control insulin-like signaling and FOXO in diapause processes, various studies have revealed that insects can produce many different insulin-like peptides (ILPs), which are important regulatory peptides in the insulin-like signaling pathway. However, not all of these peptides are involved in diapause. It is known that ILP must bind to an insulin receptor (IR) to affect its functions [19,20]. This process is highly correlated with the cellular level of protein tyrosine phosphorylation [21,22,23,24]. Further studies have shown that the large family of signaling enzymes known as protein tyrosine phosphatase 1 B (PTP1B) [25] work together with protein tyrosine kinases (PTK) to collectively modulate the cellular level of protein tyrosine phosphorylation in cells [26,27,28]. PTP1B can regulate the metabolism of locusts by regulating the amount of oxygen ions to mediate the activity of insulin signaling, whose response of PTP1B is reversible [29,30]. Forkhead transcription factors of the O class (FoxO3a)-dependent signaling pathway regulates reactive oxygen species (ROS) production by regulating NADPH oxidase 1 (nox1) activity [25,31]. These cases suggest that PTP1B and PTK are upstream modulators of the insulin-like signaling pathway (ISP) that may be involved in the regulation of diapause in locusts.

Previous studies showed that peroxidase was upregulated in diapause eggs that could clear ROS. Importantly, ROS can regulate tyrosine phosphorylation dependent signal transduction [30,32]. Here, seeking to identify the upstream modulators that control the known regulators of diapause, we examined the facultative diapause insect *Locusta migratoria* L.; we used both RNAi and an inhibitor to alter PTK and PTP1B in adult tissues. Specifically, we injected peroxiredoxin (Prx V) or RNAi Prx V under short photoperiod conditions and monitored ISP gene expression, phosphorylation of IR and IRS, the activity of nicotinamide adenine dinucleotide phosphate (NADPH) oxidase, the accumulation of reactive oxygen species (ROS), and energy metabolism. We found that PTP1B in female adults induces egg diapause under short photoperiod conditions. Further, we demonstrate that PTK in female adults inhibits egg diapause under short photoperiod conditions. Intriguingly, we also found that the antioxidant enzyme Prx V is a negative regulator of NADPH oxidizing reaction and apparently decreases ROS production and NADPH oxidase activity. Thus, our study ultimately links Prx V to the activity of both PTP1B and PTK; and these are upstream modulators controlling ISP gene expression as well as IR and IRS phosphorylation. Our results suggest a mechanism wherein diapause related proteins (PTP1B, PTK, and Prx V) of female adults can regulate offspring eggs diapause through the insulin-like signaling pathway.

## 2. Materials and Methods

### 2.1. Rearing Conditions

*L. migratoria* were reared at 28 °C, on wheat (*Triticum aestivum* L.) and the whole generation maintained from hatching through to adult maturity, either under long photoperiod (L16:D8) that induce female adults to produce nondiapausing eggs, or under short photoperiod (L10:D14) to produce eggs that enter a diapause state [32].

### 2.2. RNAi Treament

#### 2.2.1. mRNA Quantification

Sixty mature females (20 individuals per treatment, replicated three times) were randomly collected from either the long- or short- photoperiod and then dissected at low temperature (4 °C) to remove both hind legs, ovaries and fat bodies. The tissue for each body component was kept separate for each individual. Total RNA was isolated from each of the body components using TRIzol reagent (Invitrogen). After treatment with EX Taq DNase (Invitrogen Corp., Carlsbad, CA, USA), 1 μg of total RNA was used for cDNA synthesis with Takara reverse transcriptase. Relative transcript levels of *PRX5*, *PTP1B*, *PTK*, *IR*, *IRS*, *PI3K*, *Akt* from the hind legs, ovary and fat bodies were measured by quantitative RT-PCR (qRT-PCR) by using the 2(^−^ΔΔCT) method [33]. Beta-actin gene [34] served as a reference gene for normalizing the data [35]. Primer sequences including beta-actin used for qRT-PCR are listed in Table A1. We performed RT-PCR with the SYBR Premix ExTaqTM (TaKaRa, Dalian, China) on the ABI 7500 Real-Time PCR System (Applied Biosystems, Foster City, CA, USA). The reaction was performed using the following conditions: denaturation at 95 ration 60 s, followed by 40 cycles of amplification (95 °C, 15 s; 60 °C, 60 s). Three biological replication samples were used to measure gene levels by qPCR.

#### 2.2.2. RNAi

To downregulate mRNA, dsRNA for PRX V, PTP1B and PTK were synthesized using T7 RiboMAX_TM_Express RNAi System (Table A1 shows primer sequences) with the GFP gene providing the control according to the manufacturer’s protocol. Before cloning, these fragments were cross-examined using dot-plot analysis to avoid interference of the individual dsRNAs with multiple target genes. To test the efficiency of RNAi, a separate cohort of one-day-old adult females were reared under short photoperiod (L:D = 10:14) and long photoperiod conditions (L:D = 16:8), then injected with 10 μL of 1 μg/μL dsRNA solution into the ventral area between the second and third abdominal segments of female adults. After 48 h of injection, the hind leg, fat body and egg were collected, respectively, and the efficiency of RNAi-mediated depletion of each targeted mRNA was verified by qRT-PCR. There were 20 females per RNAi treatment with a total of 60 individuals for either photoperiod treatment [36].

After RNAi PRX V gene, the concentration of reactive oxygen species (ROS, O^2−^) and the enzyme activity of Nicotinamide Adenine Dinucleotide Phosphate oxidase (NADPH-OX) *in L.migratoria* were detected and calculated using enzyme linked immunosorbent assay (ELISA) with the specified manufacturer’s instructions (Collodi Biotechnology Co., Ltd., Quanzhou, China) [36].

### 2.3. Inhibitor Treatments

(i) PTP1B inhibitor PTP1B-IN-1 [37] was purchased from MedChemExpress USA. A separate cohort of one-day-old adult females were injected with 10 μL containing 300 ug PTP1B-IN-1 solution while a control cohort was injected with 10μL dimethyl sulfoxide (DMSO, AMRESCO Company, Washington State, USA) in the adult female prothorax. (ii) PTK inhibitor genistein [38] was purchased from Cayman Company (St Louis, MO, USA). One-day-old female adults were injected with 10 μL containg 50 μg gnistein solution while a control cohort was injected with 10 μL DMSO in the adult female prothorax. There were three replicates per treatment. Three days after inhibitor injection, adults were killed by 75% ethanol and the ovaries, fat bodies and 10 days-old eggs maintained at 27 °C at either long (L:D 16:8) or short (L:D 10:14) photoperiod conditions, were collected to detect (i) the relative transcript levels of *IR*, *IRS*, *PI3K* or *Akt* by using qRT-PCR and (ii) the phosphorylation level of IR, IRS, and (iii) the contents of trehalose, glycogen, triglyceride, saturated fatty acid, unsaturated fatty acid by using ELISA with the specified manufacturer’s instructions (Collodi Biotechnology Co., Ltd., Quanzhou, China) [36]. The samples (previously collected fat bodies, hind leg and ovaries of adult female locusts stored at −80 °C) were weighed and homogenized in 1 mL PBS. The homogenate was then subjected to ultrasonic treatment to further destroy the cell membrane. The homogenate was then centrifuged at 2348× *g* for 15 min. The supernatant was collected and stored in the −80 °C For further analysis.

All of the required reagents and samples, comprising a 20 mL 30 times concentrated detergent scrubbing solution, 6 mL of stop solution, 6 mL of enzyme labeling reagent, standards × 6 vials (0.5 mL × 6 vials), a micro ELISA strip plate (12 × 8 strips), 6 mL of standards diluent, 6 mL of sample diluent, 6 mL of chromogen solution A, 6 mL of chromogen solution B, two closure plate membranes and a sealed bag, were prepared and maintained properly at room temperature (25 °C) for 30 min prior to initiating further assay. The phosphorylation level of IR, IRS and the contents of trehalose, glycogen, triglyceride, saturated fatty acid and unsaturated fatty acid were detected and calculated according to the corresponding ELISA kit operation method. The detection method of the phosphorylation level of IR and IRS: (1) Take out the required slats from the aluminum foil bag after equilibration at room temperature for 20 min (the slats are precoated with insect InR, P-InR, IRS, and P-IRS antibodies), and the rest are sealed in a ziplock bag and put it back to 4 °C; (2) Set standard hole, sample hole and blank hole, and add 50 μL sample to be tested in the standard hole and sample hole, without blank hole; (3) Add 100 μL horseradish peroxidase (KRP) labeled detection antibody into the standard hole and sample hole, and seal the reaction hole with a sealing film, and bathe in water at 37 °C for 60 min; (4) Discard the filtrate, pat dry on absorbent paper, fill 350 μL into each hole, and let stand for 1 min, shake off the washing solution, pat dry on absorbent paper, and repeat 5 times; (5) Add 50 μL of Substrate A and Substrate B in the corresponding kit to each well, and incubate for 15 min in the dark at 37 °C. Add 50 μL termination solution into each hole, and measure the OD450 value of each well at the wavelength of 450 nm within 15 min.

The phosphorylation levels of INR and IRS were calculated as follows: P-InR level = (P-InR)/(InR + (P-InR)), P-IRS level = (P-IRS)/(IRS + (P-IRS)). The detection method of the content of trehalose, glycogen, triglyceride, saturated fatty acid and unsaturated fatty acid are as follows: the microplates were coated with purified trehalase, glycogen triglyceride, saturated fatty acid and unsaturated fatty acid antibody respectively to make solid-phase antibody. Trehalase glycogen triglyceride, saturated fatty acid and unsaturated fatty acid were added into the microplates coated with monoclonal antibody in turn respectively, and then combined with HRP labeled trehalase antibody to form antibody antigen enzyme labeled antibody complex. After thorough washing, TMB was added to develop the color. TMB is transformed into blue under the catalysis of HRP enzyme and yellow under the action of acid. There was a positive correlation between the color and trehalase glycogen triglyceride, saturated fatty acid and unsaturated fatty acid antibody respectively in the samples. The absorbance (OD value) was measured at 450 nm by microplate reader, and the concentration of trehalase was calculated by standard curve.

The content of trehalose, glycogen, triglyceride, saturated fatty acid and unsaturated fatty acid were calculated as follows: taking the concentration of the standard substance as the abscissa and the OD value as the ordinate, draw a standard curve and find the corresponding concentration from the standard curve according to the OD value of the sample; then multiply by the dilution factor; or use the concentration of the standard substance to calculate the linear regression equation of the standard curve with the OD value, substitute the OD value of the sample into the equation, calculate the sample concentration and multiply it by the dilution The multiple is the actual concentration of the sample.

### 2.4. Prx V Protein Purification and Injection

Prx V protein was expressed and purified by Pichia Pastoris System [39]. Pichia pastoris GS115 and yeast expression vector pPIC9K provided by Hongxun Company in Suzhou, China. The cultured saccharomycetes were broken by superwave cell crusher (4500 rpm, 4 s, 6 s interval, 300 cycles in total) (JY 96-IIN, NINGBO SCIENTZ Biotechnology Co., Ltd., Ningbo, China). The buffer solution for saccharomycetes were 20 mM Tris, 500 mM NaCl, 20 mM imidazole, pH = 8.0. The broken saccharomycetes were placed in a high-speed freezing centrifuge (Sigma) (4 °C, 12000 rpm) for 20 min, and the supernatant was collected for nickel column purification (Ni-IMAC, 5 mL) and the purified protein was centrifuged in a 4 mL 10 KD protein ultrafiltration tube and the concentration was 2.675 mg/mL. One-day-old adult females were injected with either 26.75 μg in 10 μL Prx V solution or 10μL protein solvent in the adult female prothorax (20 mM PBS, 500 mM NaCl) solution which served as the control. Thereafter, adults were maintained on wheat and allowed to oviposit in sand; eggs were collected and held to determine the incidence of diapause. There were three replicates per treatment with 30 eggs in each duplication.

### 2.5. Collection of Eggs

All female adults of the above three treatments and corresponding controls from both long (L:D 16:8) and short (L:D 10:14) photoperiod treatments were held in rearing chamber with a sand base through to maturity and oviposition. As reproduction can only occur after mating, males were added to the cages once females reached maturity. Egg masses were collected by hand from the sand floor and eggs removed, sterilized using 1% bleach and placed on filter paper in Petri dishes maintained at 27 °C under a L:D 12:12 photoperiod for 15 days until emergence had ceased. For the *Locusta migratoria*, only the fertilized egg can hatch into a nymph. The number of hatching eggs was recorded as D1. In order to terminate diapause, the remaining eggs were stored at 4 °C for one month, and then incubated at 32 °C, and the number of hatching larvae is counted as D2. The remaining unhatched eggs are unfertilized or dead eggs. The filter paper was periodically moistened with distilled water to prevent egg desiccation [40]. There were three duplications per treatment with 30 eggs in each duplication.

### 2.6. Statistical Analysis

The percent of eggs entering diapause was determined using the formula Eggs’ diapause incidence = the formula Eggs’ diapause incidence = D2/(D1 + D2), with unfertilized or dead eggs ignored. Analysis was performed using Student’s *t*-test by SAS (version 8.0, SAS Inc., Chicago, IL, USA) statistical software package.

## 3. Results

### 3.1. Female Adult PTP1B Promotes Egg Diapause under the Condition of Diapause Induction

#### 3.1.1. PTP1B RNAi Increased the Expression Level of the ISP Genes and Inhibted Diapause under Short Photoperiod Conditions

Treatment of *Locusta migratoria* L. female adults under short photoperiod conditions resulted in a decrease in PTP1B gene expression in fat bodies, ovaries and hind legs, by 90.2%, 31.0% and 95.7%, respectively (Figure 1A). In the fat bodies, ovaries and hind legs, the relative mRNA expression level of IR, IRS, PI3K and AKT genes in the insulin-like signaling pathway (ISP) was significantly increased after RNAi PTP1B as compared to the control (Figure 1B–E). Further, treatment with RNAi PTP1B decreased the diapause incidence in offspring eggs by 23.7% (*p* = 0.0011) (Figure 1F). However, all the eggs laid by RNAi PTP1B treated adults that were reared under long photoperiod conditions hatched successfully; yet RNAi PTP1B gene treatment under long photoperiod condition did result in a significant increase in expression of the IR, IRS, PI3K and AKT genes in the fat bodies, ovaries and hind legs (Appendix A
Figure A1). Thus, the phosphatase gene PTP1B functions to negatively regulate the expression of the ISP genes under short photoperiod conditions in migratory locusts.

#### 3.1.2. PTP1B Activity on Egg Diapause

Following injection of the adult female prothorax with the known PTP1B inhibitor PTP1B-IN-1 [37] the transcription levels of *IR*, *IRS*, *PI3K* and *AKT* genes in the fat bodies and ovaries (Figure 2A–D), and the phosphorylation levels of IR and IRS in the 10-days-old eggs, each increased significantly (Figure 2E). The diapause incidence was significantly reduced (by 21.4%) after PTP1B-IN-1 treatment under short photoperiod conditions (*p* = 0.0022) (Figure 2F). However, all the eggs laid by adults treated with PTP1B-IN-1 and reared under long photoperiod conditions hatched successfully; this treatment increased the expression of *IR*, *IRS*, *PI3K* and *AKT* genes and increased the phosphorylation level of IR and IRS under long photoperiod conditions (Figure A2). These results further confirm that the PTP1B phosphatase functions in *Locusta migratoria* L. to negatively regulate the expression of *ISP* genes under short photoperiod conditions. Notably, we also found that this injection of PTP1B-IN-1 into adult females grown under short photoperiod conditions altered the energy metabolism of the10-days-old eggs; specifically, this treatment significantly decreased the levels of glycogen (*p* = 0.0072), triglycerides (*p* = 0.0019) and unsaturated fatty acids (*p* = 0.0094) but significantly increased the levels of trehalose (*p* < 0.001) and saturated fatty acids (*p* = 0.0001) (Figure 2G,H). Collectively, these results showed that the PTP1B inhibitor PTP1B-IN-1 can inhibit egg diapause, promote ISP gene expression, increase phosphorylation of IR and IRS and alter the energy metabolism of eggs laid by adults reared in short photoperiod conditions.

### 3.2. Female Adult PTK Inhibts Egg Diapause under the Condition of Diapause Induction

#### 3.2.1. PTK RNAi Decreased the Expression Level of the ISP Genes and Promoting Diapause under Short Photoperiod Conditions

After RNAi *PTK* treatment of females under short photoperiod conditions, the expression of the *PTK* gene in fat bodies, ovaries and hind legs decreased by 91.2%, 65.5% and 98.5% respectively (Figure 3A). In the fat bodies, ovaries and hind legs, the expression of *IR*, *IRS*, *PI3K* and *AKT* of the insulin-like signaling pathway were significantly decreased after RNAi *PTK* gene treatment (Figure 3B–E), results showing that the *ISP* is inhibited upon RNAi *PTK* gene under short photoperiod conditions. Further, compared with the control group, the diapause incidence of the offspring eggs increased by 28.7% after treatment with RNAi *PTK* gene (*p* < 0.0011) (Figure 3F). Thus, PTK functions as a positive regulator of the expression of *ISP* genes in short photoperiod conditions. In contrast, all the eggs laid by adults treated with RNAi PTK and reared under long photoperiod conditions hatched successfully, but this treatment significantly decreased the expression of the *IR*, *IRS*, *PI3K* and *AKT* genes in fat bodies, ovaries and hind legs (Figure A3).

#### 3.2.2. PTK Inhibit on Egg Diapause

After female adults were injected with PTK inhibitor genistein [38] under short photoperiod conditions, the transcription levels of *IR*, *IRS*, *PI3K* and *AKT* genes in the fat bodies and ovaries (Figure 4A–D), as well as the phosphorylation levels of IR and IRS in the 10-days-old eggs, decreased significantly upon genistein treatment under short photoperiod conditions (Figure 4E). Moreover, the diapause incidence increased significantly (by 27.8%) upon genistein treatment under short photoperiod conditions (*p* = 0.0012) (Figure 4F). These results further support that PTK is a positive regulator of ISP genes under short photoperiod conditions. In contrast, the eggs laid by genistein-treated adults reared under long photoperiod conditions were all able to hatch but exhibited significantly decreased expression of the *IR*, *IRS*, *PI3K* and *AKT* genes and significantly reduced the phosphorylation levels of IR and IRS (Figure A4). Moreover, the injection of genistein into adult females grown under short photoperiod conditions altered the energy metabolism of the 10-days-old eggs. This treatment significantly increased glycogen (*p* = 0.0003), triglyceride (*p* = 0.0012), and unsaturated fatty acid (*p* = 0.0004) content but significantly decreased trehalose (*p* < 0.001) and saturated fatty acid (*p* = 0.0002) content (Figure 4G,H). These results showed that treatment of adults reared under short photoperiod conditions with the PTK inhibitor genistein can induce diapause in offspring eggs, inhibit the expression of ISP genes, increase IR and IRS phosphorylation and alter the energy metabolism of eggs.

### 3.3. Female Adult Prx V Promoted Egg Diapause under the Condition of Diapause Induction

#### 3.3.1. RNAi of Prx V Inhibits Eggs’ Diapause Incidence

Treatment of female adults with RNAi targeting the *Prx V* gene under short photoperiod decreased *Prx Vexpression* in fat bodies, ovaries and hind legs by 99.7%, 99.0% and 84.4%, respectively (Figure 5A). In fat bodies, ovaries and hind legs, the expression of the *ISP* genes *IR*, *IRS*, *PI3K*, and *Akt* increased significantly after RNAi of *Prx V* (Figure 5B–E). Compared with untreated controls, the diapause incidence of the *Prx V* RNAi offspring eggs decreased by 46.0% (*p* = 0.0015) (Figure 5F), implicating Prx V as a negative regulatory of *ISP* gene expression and diapause. However, the eggs of adults reared in long photoperiod condition all hatched after *Prx V* RNAi treatment, but this treatment did variously affect the expression trends for the *ISP* genes *IR*, *IRS*, *PI3K* and *Akt* genes in different issues (Figure A5).

Strikingly, we also noted that RNAi of P*rx V* significantly increased the levels of reactive oxygen species (ROS) and NADPH redox (NADPH-OX) in fat bodies, ovaries and hind legs, and this was true under both short and long photoperiod conditions (Figure 5G,H). Notably, although the ROS and NADPH-OX activity increased upon RNAi of Prx V under long photoperiod conditions, all the eggs laid by these adults had hatched successfully (Figure A5). These results establish that although Prx V can regulate ROS and NADPH-OX levels in both short and long photoperiod conditions, it only affects locust diapause under short photoperiod conditions.

#### 3.3.2. Injected Prx V Protein Induces Eggs’ Diapause

We also purified the Prx V protein and injected it into the prothorax of adult females. Compared to untreated controls, the diapause incidence of the offspring eggs increased by 20.5% upon Prx V protein injected in female adults reared in under short photoperiod conditions (*p* = 0.0283, Figure 5I); all the eggs laid by for Prx V-protein-injected females reared under long photoperiod conditions hatched successfully (Figure 5I). These results show that injecting the Prx V protein can induce eggs’ diapause under short photoperiod conditions.

### 3.4. Antagonism between PTP1B and PTK Mediates in the ISP of Adults Regulates Egg Diapause in Migratory Locust

The aforementioned results establish that PTP1B and PTK can regulate the transcription the ISP genes *IR*, *IRS*, *PI3K* and *AKT* and regulate the phosphorylation of *IR* and *IRS* under short photoperiod conditions. Locust diapause is induced by PTP1B (Figure 1 and Figure 2) but inhibited by PTK (Figure 3 and Figure 4). RNAi of the *Prx V* gene and injection of Prx V protein into female adults reared under short photoperiod conditions revealed that Prx V decreases ROS production and NADPH-OX activity (Figure 5), inactivates PTK and decreases the expression ISP genes including *IR*, *IRS*, *PI3K* and *AKT* and thus acts as an inducer of locust diapause. Thus, we propose that Prx V can regulate the activity of PTP1B and PTK by regulating the extent of NADPH oxidation and thereby controls the ISP signaling that mediates the diapause of locust eggs (Figure 6A,B).

## 4. Discussion

It is well-established that the diapause process comprises three main steps: environmental information transmission, the actual regulation of diapause and development after diapause [42] (pp. 94–101). We previously conducted an integrated “omics” study that examined the environmental information transmission step of locust diapause and which revealed the differential expression of the Prx V protein in diapause vs. nondiapause eggs [32,43]. In the present study, we show that the Prx V protein can inhibit ROS and can activate PTP1B to induce locust diapause. However, locust diapause is a complicated process, and there are still additional diapause-related proteins waiting to be discovered [5]. We found that PTP1B and PTK not only regulate the level of tyrosine phosphorylation of various proteins in cells but also can affect the synthesis of energy substances in embryos. These findings are in line with previous studies reporting that the RNAi of *PTP1B* upregulates the activity of insulin-controlled fatty acid synthase in the prediapause stage [44,45,46] and reporting that embryos are able to synthesize cryoprotectants and store energy for later developmental stages [1,47]. We speculate that the NADPH oxidation activity likely determines the balance of the expression and activity between PTP1B and PTK. Consider that NADPH oxidation can promote the production of ROS [48] and that the intracellular ROS concentration is closely related with insect life span [49]. For instance, overexpression of NADH dehydrogenase in young adult brains has been shown to significantly extend the life span of fruit flies and to promote adult diapause [50]. In this paper, we found that RNAi-mediated knockdown of *Prx V* (encoding a peroxiredoxin) in mature females ultimately decreased the extent of diapause in offspring eggs. We also found that increase in ROS accumulation corresponds with activation of PTK, which then inhibits egg diapause. After injection with Prx V protein, the diapause incidence in offspring eggs increased, which suggested the possibility that decreased ROS accumulation may activate PTP1B and thereby subsequently promote egg diapause. This conclusion would be completely opposed to the previously reported result that ROS extend insect life span [50] but highly consistent with the conventional understanding that increased intracellular ROS accelerants aging [51,52]. It is likely that these differences relate to the divergent diapause strategies employed by various species [5].

The regulation of insect diapause is a very complex process. Although the present study reveals a contribution from Prx V in regulating the ROS burst that is known to trigger ISP gene transcription, the regulation steps prior to Prx V have not yet been determined. We know that both diapause hormone (DH) and insulin-like peptide (ILP) mediate diapause in insects [53,54]. Generally, DH has been found to promote diapause [47], while ILP has been found to inhibit diapause [17,54]. In other words, DH apparently functions under short photoperiod conditions whereas ILP functions under long photoperiod conditions [55,56]. However, we found ILP has been inhibited under short photoperiod conditions, but locust embryos can still develop for a short time during the diapause induction stage. Thus, there is likely a mechanism that mediates the accumulation of DH and ILP during the diapause induction stage under short photoperiods. We speculate that this mechanism may involve G protein coupled receptor (GPCRs), based on the following considerations: (i) Diapause hormone receptor (DHR), an important receptor in the DH regulation, belongs to the GPCR family [47]; while insulin-like receptor (IR), an important receptor in the ILP regulation, belongs to the RTK family [19]. (ii) GPCR agonist stimulation of GPCR triggers the activation of several second messengers such as the reactive oxygen species (ROS), which in turn, promote tyrosine phosphorylation and subsequent activation of RTKs [41]. The possible regulatory process can be shown that GPCR, through activation of ROS, may mediate the balance between PTP1B and PTK and influence the diapause incidence of migratory locusts through the insulin-like signaling pathway during the diapause induction stage (Figure 6C).

## 5. Conclusions

The aforementioned results establish that DH may take a dominant role in helping locust eggs to enter diapause under short photoperiod conditions. While under long photoperiod conditions, we speculate that ILP functions during the whole developmental stage and inhibits other factors. Prx V can regulate the activity of PTP1B and PTK by regulating the extent of NADPH oxidation and thereby controls the ISP signaling that mediates the diapause of locust eggs. This could help explain our finding that all eggs hatched under long photoperiod conditions, and the diapause incidence have been affected only under short photoperiod conditions with the interruption of PTP1B, PTK and Prx V. Taken together, these results indicate that PTP1B, PTK and Prx V are upstream modulators that regulate diapause in eggs via the insulin signaling pathway. Furthermore, these findings have revealed a possible bridge connecting diapause hormone signaling to the insulin-like signaling pathway.

## Figures and Tables

**Figure 1 insects-12-00253-f001:**
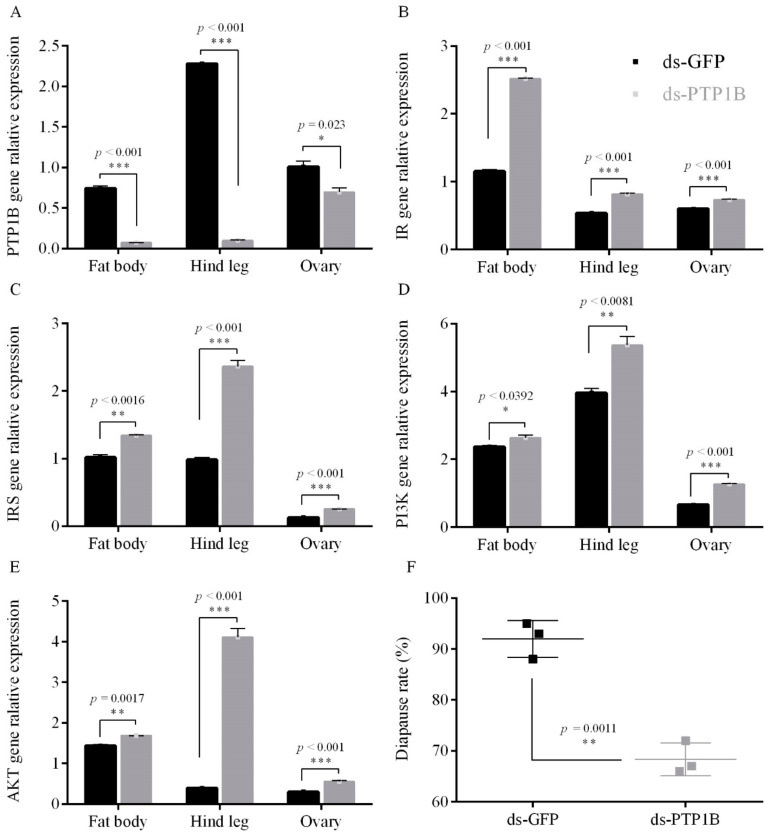
Gene relative expression and egg diapause incidence after injection of adults with RNAi *PTP1B* compared to injection with GFP (control) under short photoperiod conditions (L10: D14) for (**A**) *PTP1B* gene, (**B**) *IR* gene, (**C**) *IRS* gene, (**D**) *PI3K* gene, (**E**) *AKT* gene and (**F**) egg diapause incidence. Data are the mean ± SEM from three independent replicates, the gray bar indicates RNAi treatments, while the black bar indicates GFP controls (* *p* < 0.05, ** *p* < 0.01, and *** *p* < 0.001 vs. GFP controls, Student’s *t*-test).

**Figure 2 insects-12-00253-f002:**
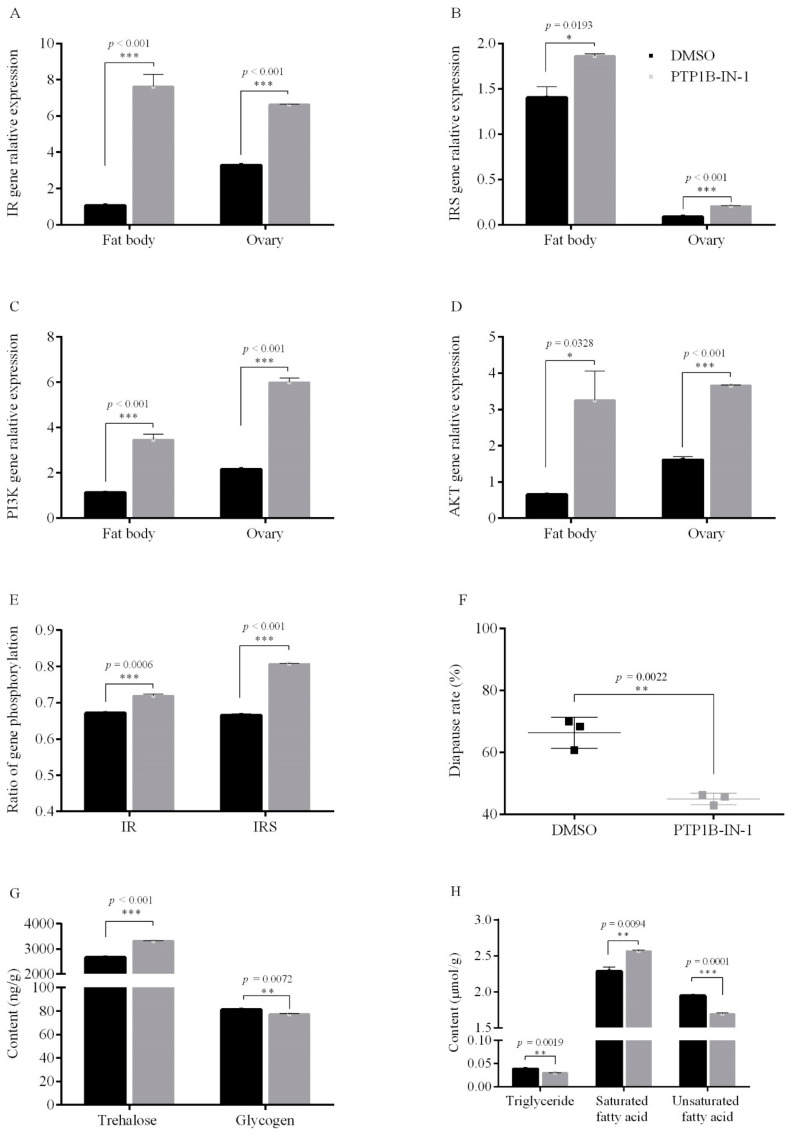
Gene relative expression, phosphorylation level, energy metabolism and egg diapause incidence after injection of adults with PTP1B inhibitor PTP1B-IN-1 compared to DMSO (control) under short photoperiod conditions (L10:D14). (**A**) *IR* gene, (**B**) *IRS* gene, (**C**) *PI3K* gene, (**D**) *AKT* gene, in the fat bodies and ovaries, (**E**) phosphorylation levels of IR and IRS in the 10-days-old eggs, (**F**) egg diapause incidence, (**G**,**H**) energy metabolism in the 10-days-old eggs. Data are the mean ± SEM from three independent replicates, the gray bar indicates PTP1B inhibitor treatments, while the black bar indicates DMSO controls (* *p* < 0.05, ** *p* < 0.01, and *** *p* < 0.001 vs. DMSO controls, Student’s *t*-test).

**Figure 3 insects-12-00253-f003:**
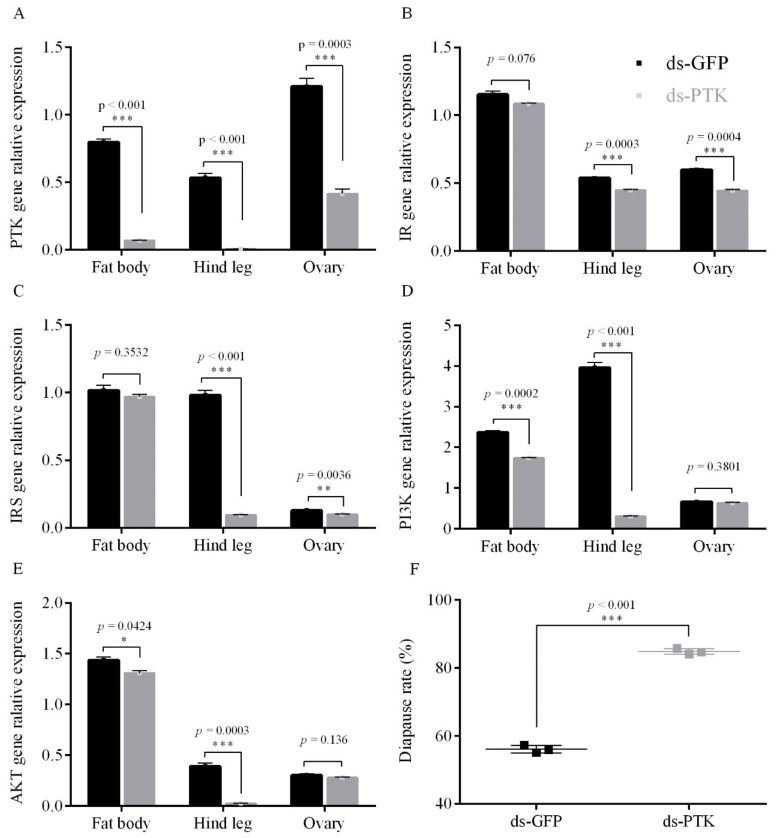
Gene relative expression and egg diapause incidence after injection of adults with RNAi PTK compared to GFP (control) under short photoperiod conditions (L10:D14) for (**A**) *PTK* gene, (**B**) *IR* gene, (**C**) *IRS* gene, (**D**) *PI3K* gene, (**E**) *AKT* gene and (**F**) egg diapause incidence. Data are the mean ± SEM from three independent replicates, the gray bar indicates RNAi treatments, while the black bar indicates GFP controls (* *p* < 0.05, ** *p* < 0.01, and *** *p* < 0.001 vs. GFP controls, Student’s *t*-test).

**Figure 4 insects-12-00253-f004:**
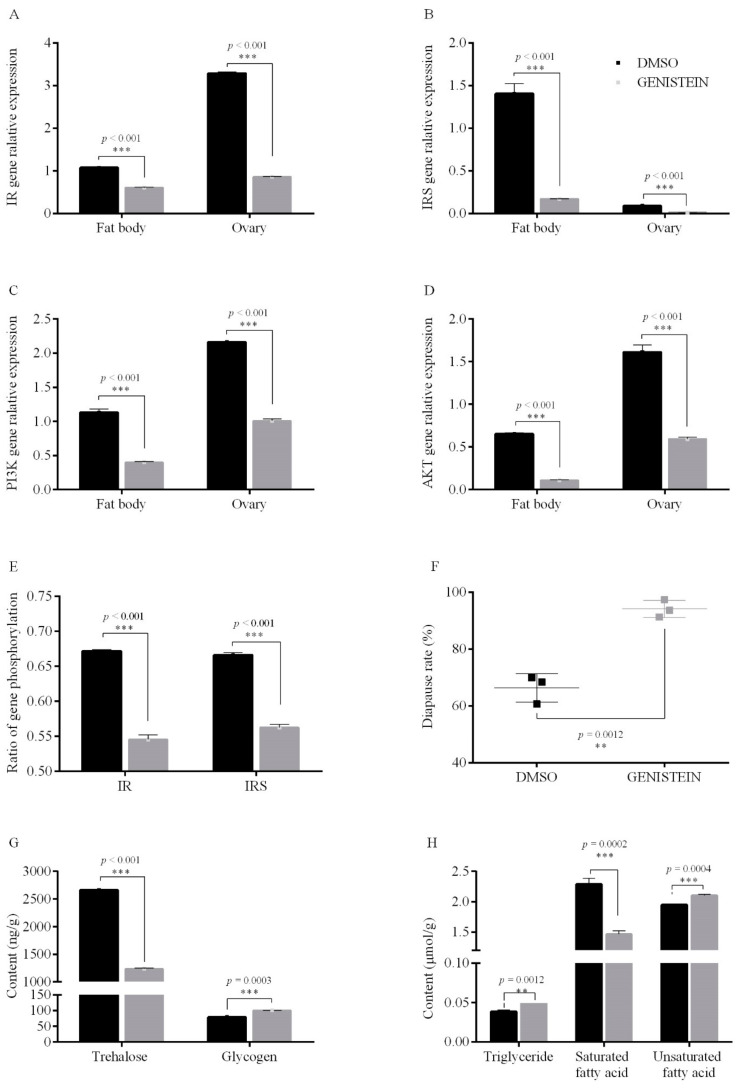
Gene relative expression, phosphorylation level, energy metabolism and egg diapause incidence after injection of adults with the PTK inhibitor genistein compared to DMSO (control) under short photoperiod conditions (L10: D14). (**A**) *IR* gene, (**B**) *IR*S gene, (**C**) *PI3K* gene, (**D**) *AKT* gene in the fat bodies and ovaries, (**E**) phosphorylation levels of IR and IRS in the 10-day-old eggs, (**F**) egg diapause incidence, (**G**,**H**) energy metabolism in the 10-day-old eggs. Data are the mean ± SEM from three independent replicates, the gray bar indicates PTK inhibitor treatments, while the black bar indicates DMSO controls (** *p* < 0.01 and *** *p* < 0.001 vs. DMSO controls as assessed, Student’s *t*-test).

**Figure 5 insects-12-00253-f005:**
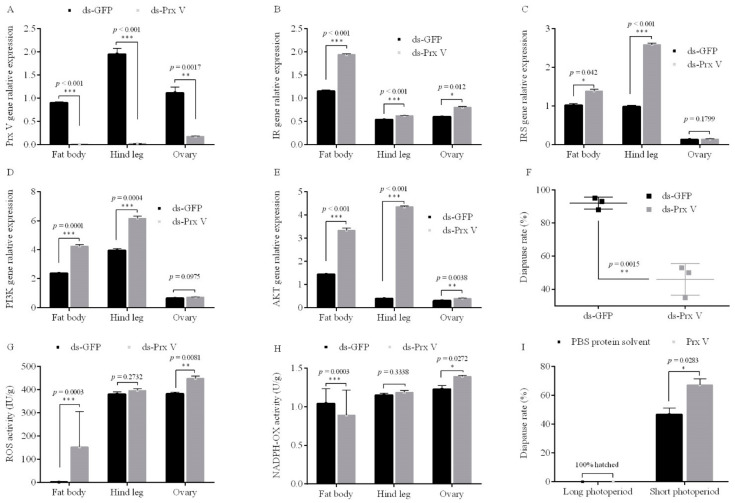
Gene relative expression, reactive oxygen species (ROS) and NADPH redox (NADPH-OX) changes in fat bodies, ovaries and hind legs, egg diapause incidence after injection of adults with the RNAi *Prx V* gene compared to *GFP* under short photoperiod conditions (L10:D14) (control), and egg diapause incidence after injection of adults with the Prx V protein compared to PBS protein solvent (control) under both long photoperiod (L16: D8) and short photoperiod conditions. (**A**) *Prx V* gene (**B**) *IR* gene, (**C**) *IRS* gene, (**D**) *PI3K* gene, (**E**) *AKT* gene relative expression, (**F**) egg diapause incidence, (**G**)ROS activity, (**H**) NADPH-OX activity after injection of adults with the RNAi *Prx V* gene compared to GFP (control). Data are the mean ± SEM from three independent replicates, the gray bar indicates RNAi treatments, while the black bar indicates GFP controls (* *p* < 0.05, ** *p* < 0.01, and *** *p* < 0.001 vs. GFP controls, Student’s *t*-test). (**I**) Egg diapause incidence after injection of adults with the Prx V protein compared to PBS protein solvent (control) under both long photoperiod and short photoperiod conditions. Data are the mean ± SEM from three independent replicates, the gray bar indicates Prx V protein treatment, while the black bar indicates PBS controls (* *p* < 0.05 vs. PBS protein solvent controls as assessed, Student’s *t*-test).

**Figure 6 insects-12-00253-f006:**
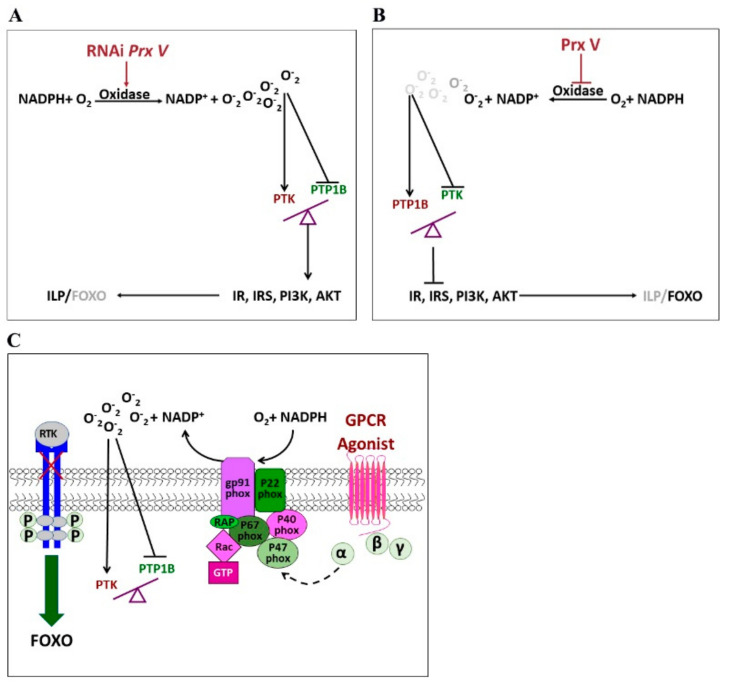
G protein coupled receptor (GPCR) through activation of ROS mediates the balance between PTP1B and PTK, influencing the diapause incidence of migratory locusts through the insulin-like signaling pathway. (**A**) and (**B**) show that antagonism between PTP1B and PTK mediates adults’ insulin-like signal regulating eggs’ diapause of migratory locusts under short photoperiod conditions. Generally, (**A**) RNAi *Prx V* gene of female adults, will promotes ROS production and NADPH-OX activity, then activates PTK, and increases the relative expression of important genes in the ISP, including *IR*, *IRS*, *PI3K*, and *AKT* gene under short photoperiod, finally promoting locust development. While (**B**) infection of the Prx V protein to female adults will inhibit ROS production and NADPH-OX activity, then activates PTP1B, and decreases the relative expression of important genes in the ISP, including IR, IRS, PI3K and AKT under short photoperiod, finally induce locust diapause. (**C**) Proposed model: the possible new mechanism wherein GPCR regulation of locust diapause. GPCR agonist (the role contrary with the Prx V) can combine the ligand p47 phox (a ligand of oxidase) to mediates the NADPH oxidation reaction [41], perhaps controlling the balance between PTP1B and PTK in the cell, then induces the PTK expression and increases expression of the *IR, IRS*, *PI3K* and *AKT* gene, also promotes the phosphorylation levels of IR, IRS in the insulin-like signaling pathway and finally promotes the embryo developmental biochemical processes under short photoperiod conditions.

## Data Availability

Figure A1: Gene relative expression and egg hatching incidence after injection of adults with RNAi PTP1B compared to injection with GFP (control) under long photoperiod conditions (L16:D8). Figure A2: Gene relative expression, phosphorylation level, energy metabolism and egg hatching incidence after injection of adults with PTP1B inhibitor PTP1B-IN-1 compared to DMSO (control) under long photoperiod conditions (L16:D8). Figure A3: Gene relative expression and egg hatching incidence after injection of adults with RNAi PTK compared to GFP (control) under long photoperiod conditions (L16:D8). Figure A4: Gene relative expression, phosphorylation level, energy metabolism and egg hatching incidence after injection of adults with the PTK inhibitor genistein compared to DMSO (control) under long photoperiod conditions (L16:D8). Figure A5: Gene relative expression, reactive oxygen species (ROS), and NADPH redox (NADPH-OX) changes in fat bodies, ovaries, and hind legs and egg hatching incidence after injection of adults with the RNAi Prx V gene compared to GFP (control) under long photoperiod conditions (L16:D8). Table A1. All gene sequences involved in this paper.

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
