# Peer review of "Antagonism between PTP1B and PTK Mediates Adults’ Insulin-Like Signaling Regulation of Egg Diapause in the Migratory Locust"

_insects, 2021, doi:10.3390/insects12030253_

Round 1

Reviewer 1 Report

The manuscript by Li et al describes the role of two proteins, PTP1B and PTK in regulating insulin-like signaling in the insect Locusta migratoria. In their study, the researchers used RNAi-mediated knockdown of key proteins to demonstrate how the insulin-like signaling pathway can control induction of diapause in locusts exposed to short photoperiod conditions. They also used chemical inhibitors to support their RNAi-based observations, and concluded that PTP1B can induce diapause while PTK inhibits it. They also produced an antioxidant protein, Prx V, in a yeast expression system, and demonstrated that when injected into the locusts, diapause was induced in the locust embryos. They propose an intriguing model that describes how production of reactive oxygen species may modulate the balance of the two opposing regulatory proteins’ activities to control when embryonic induction can occur.

The study is well conducted, and the manuscript presents an extensive collection of data to help explain how induction of diapause may be controlled at an intracellular signaling level. These findings will contribute significantly to future work that explores the complex network of signals, from environmental cues (shortening daylight), hormonal signaling (diapause hormone vs insulin-like peptides), and effector genes and proteins that trigger changes in cellular metabolic states to induce diapause. I found the results to be clearly presented, and many of the conclusions logical. However, I have some questions that I believe should be considered before final publication.

  1. Line 130-1. “…the enzyme activity of reactive oxygen species (ROS)…..were detected” is not providing enough detail of how ROS (not an enzyme!) were measured. What reactive oxygen species were measured, and how was that accomplished? How were tissues processed, etc.? A reference [36] is cited, but this present manuscript should stand alone and provide enough details to understand how the results were obtained.
  2. Lines 146-149. The authors indicate that they measured phosphorylation levels of IR and IRS, and protein levels of trehalose, glycogen, triglycerides, and fatty acids using ELISA. First, the latter items are not proteins, and secondly, the reference [36] they cited failed to provide details on how these molecules were isolated and measured. Please provide experimental details.
  3. Can the authors comment on why perturbations in transcript abundance were almost always stronger in the hind leg compared to the fat body or ovary? Could the differences be associated with using actin as a reference gene for the qRT-PCR, as actin would be highly expressed in leg muscle, whereas it would be a lower abundance in fat body and ovaries? Can the authors comment on using just one reference gene in their qRT-PCR analyses, as the convention is now to use at least two?
  4. For the experiments involving injection of locusts with yeast-expressed Prx V protein, negative control insects were injected with buffer. Can the authors comment why they did not use a non-specific protein or a non-functional Prx V protein (heat inactivated?) as a negative control, to provide clear evidence that the Prx V protein was inducing the effects on the locusts?
  5. The model in Figure 6 should be called a “proposed” model, as the authors did not examine the role of a GPCR in the regulatory pathway, and the interaction of the cell surface receptors (the GPCR and the RTK) is speculation. The Discussion describing this model (lines 410-420) is intriguing, but is there evidence that ROS differentially impacts PTK and PTP1B activity? In this study, change in gene expression were evaluated, but no measures of enzyme activity were determined.

Author Response

We thank the anonymous reviewer for his/her encouragement and marking ‘yes’ for most of the qualitative attributes mentioned above. M&M section has been improved to the best of our ability and understanding. For the conclusions, we accepted the reviewer's suggestion revised in the Figure 6. 

As for the response, please see the attachment.

Reviewer 2 Report

The manuscript by Shuang Li et al about “Antagonism bewtee PTP1B abd PTK mediating adults insulin like signaling regulation of Egg diapause in the migratory locust”  submitted for publication in Insects, is experimentally sound and clearly written in well understandable phrases. The experimental data and results, based upon the use of selective RNAi knockdown and validated by subsequent use of selective inhibitors are presented in a  correct set of 5 figures which are well explained in the text and have correct annotations.

Only minor corrections need attention and should be addressed appropriately.

Line 40 Bombyx mori needs to be written in in italics

Line 96 oC should be replaced by °C

Line 119 reads incorrect. Suggestion “To downregulate mRNA, dsRNA for PRC V, PTP1B and PTK were synthetized…”

Line 130-134 Enzyme activity of reactive oxygen species???? And of NADPH-OX were detected using ELISA???Elisa refers to antibody based assay which quantifies antigen content. ROS  can be measured using fluorescent plate reader assay and idem for measuring NADPH-OX activity using correct substrate

Line 137-138 duplication of word “solution” is incorrect, idem line 141!

Line 144 10 days-old eggs of animals maintained at 27°C at either long     or short    photoperiod

Line 147-148 Is ELISA indeed the method used for measuring trehalose, glycogen, triglyceride satutaed and unsaturated fatty acids????? Is this not by a method based upon use of fluorescence and plate reader at    nm?????

Line 172 Locusta migratoria in italics! And even better omit sentence at line 173)174 “ Only the fertilized eggs can hatch into locust while the eggs that can’t hatch into larvae are unfertilized or dead [40] cf identical info at line 177

Line 187 replace inhibiting by inhibited

In the discussion section I miss some explanation about why the RNAi’s described do not have any effect in the situation when locust’s are reared in LD condtion? Some upstream photoperiod reguletd factor/sensor is still missing in order of completing the pictures as summarized in FIG 6

Author Response

We thank the anonymous reviewer for his/her encouragement and marking ‘yes’ for most of the qualitative attributes mentioned above. M&M section has been improved to the best of our ability and understanding.

As for the response to the reviewer’s comments , Please see the attachment.

Round 2

Reviewer 1 Report

The authors have addressed most of the issues from the first review.

However, the description of how 1) the phosphorylation level of IR and IRS, 2) the content of trehalose, 3) glycogen, 4) triglyceride, and 5) saturated fatty acid, unsaturated fatty acid were detected and calculated according to using ELISA  methods needs more clarification. Which specific ELISA kits were used to make those measurements? Are those kits capable of distinguishing the subtle differences in phosphorylation states of the IR/IRS complex, and can they clearly distinguish saturated and unsaturated fatty acids? Specific details are needed, so that others can use similar methods in their own studies.

Author Response

Dear reviewer.

 We really appreciate the keen interest, fruitful suggestions, appreciation and encouragement by the honorable reviewer. To the best of our knowledge, the revised version in the light of the comments of the reviewer has been greatly improved.
